# Effects of *sn*-2 Palmitic Triacylglycerols and the Ratio of OPL to OPO in Human Milk Fat Substitute on Metabolic Regulation in Sprague-Dawley Rats

**DOI:** 10.3390/nu16091299

**Published:** 2024-04-26

**Authors:** Lin Zhu, Shuaizhen Fang, Yaqiong Zhang, Xiangjun Sun, Puyu Yang, Weiying Lu, Liangli Yu

**Affiliations:** 1Institute of Food and Nutraceutical Science, School of Agriculture and Biology, Shanghai Jiao Tong University, Shanghai 200240, China; julia_1113@sjtu.edu.cn (L.Z.); fangsz9918@sjtu.edu.cn (S.F.); xjsun@sjtu.edu.cn (X.S.); yangpuyuallen@sjtu.edu.cn (P.Y.); weiying.lu@sjtu.edu.cn (W.L.); 2Department of Nutrition and Food Science, University of Maryland, College Park, MD 20742, USA; lyu5@umd.edu

**Keywords:** human milk fat substitute, *sn*-2 palmitic triacylglycerols, metabolomics, lipidomics

## Abstract

In this study, the influence of total *sn*-2 palmitic triacylglycerols (TAGs) and ratio of 1-oleoyl-2-palmitoyl-3-linoleoylglycerol (OPL) to 1,3-dioleoyl-2-palmitoylglycerol (OPO) in human milk fat substitute (HMFS) on the metabolic changes were investigated in Sprague–Dawley rats. Metabolomics and lipidomics profiling analysis indicated that increasing the total *sn*-2 palmitic TAGs and OPL to OPO ratio in HMFS could significantly influence glycine, serine and threonine metabolism, glycerophospholipid metabolism, glycerolipid metabolism, sphingolipid metabolism, bile acid biosynthesis, and taurine and hypotaurine metabolism pathways in rats after 4 weeks of feeding, which were mainly related to lipid, bile acid and energy metabolism. Meanwhile, the up-regulation of taurine, L-tryptophan, and L-cysteine, and down-regulations of lysoPC (18:0) and hypoxanthine would contribute to the reduction in inflammatory response and oxidative stress, and improvement of immunity function in rats. In addition, analysis of targeted biochemical factors also revealed that HMFS-fed rats had significantly increased levels of anti-inflammatory factor (IL-4), immunoglobulin A (IgA), superoxide dismutase (SOD), and glutathione peroxidase (GSH-px), and decreased levels of pro-inflammatory factors (IL-6 and TNF-α) and malondialdehyde (MDA), compared with those of the control fat-fed rats. Collectively, these observations present new in vivo nutritional evidence for the metabolic regulatory effects of the TAG structure and composition of human milk fat substitutes on the host.

## 1. Introduction

Human breast milk has been recognized as the most ideal food source for infants. Breast milk fat contributes approximately 50% of the total energy needed for infant growth and development, which is well known for its desirable absorption efficiency, immunity, and intestinal benefits [1,2]. Meanwhile, triacylglycerols (TAGs) in breast milk fat have a unique stereo-distribution and composition. About 70% of palmitic acid (PA) is esterified at the *sn*-2 position and forms *sn*-2 palmitic TAGs [3]. In addition, two main types of *sn*-2 palmitic TAGs in breast milk fat are 1,3-dioleoyl-2-palmitoyl-glycerol (OPO) and 1-oleoyl-2-palmitoyl-3-linoleoylglycerol (OPL), and their mass ratio differs in the breast milk of different ethnic groups. The OPL to OPO mass ratio was less than 1 in Western breast milk and approximately 1.35 in Chinese breast milk [4,5].

Owing to the positive nutritional effects of breast milk fat, *sn*-2 palmitic TAGs have been synthesized and supplemented into infant formula as the human milk fat substitute (HMFS) to mimic the nutritional and functional characteristics as close as possible to those of breast milk fat [6,7]. Meanwhile, our recent studies have found that the total *sn*-2 palmitic TAGs and its TAG composition in HMFS play a synergistic role in the utilization of nutrients (lipids and minerals) and energy, serum lipid, and bile acid profiles in Sprague–Dawley (SD) rats. Increasing the total *sn*-2 palmitic TAGs and OPL to OPO mass ratio could decrease body weight gain and lipid accumulation in liver and perirenal adipose tissue, and improve serum lipid parameters in high-fat fed rats [8,9]. All of these observations indicate the influence of total *sn*-2 palmitic TAGs and their TAG composition on metabolic alternations in rats. However, to the best of our knowledge, there are few studies dealing with the specific changes in metabolic pathways and related metabolites.

In addition, Guo et al. confirmed that the feces of infants fed with high levels of *sn*-2 palmitic TAGs were enriched with beneficial fecal metabolites such as amino acids and fatty acids, whose potential biological functions include inhibiting inflammation and improving immunity, compared to those of infants fed the formula using regular vegetable oil [10]. Several in vivo studies have also shown that *sn*-2 palmitic TAGs have some immunity benefits. Chen et al. found that the OPO-supplemented diet could regulate the levels of serum immunity-related factors and enhance the immune function in mice [7]. According to the observations of Wei et al., mice fed with a high-level OPL diet (consisting of 43% OPL and 57% soybean oil) had an improved immunity function compared to mice fed with soybean oil only, which might be related to the increased short-chain fatty acid-producing intestinal bacteria [11]. Meanwhile, it is known that inflammatory responses are modulated by the host immune system [12], while oxidative stress, an excessive accumulation of reactive oxygen species, could severely interfere with the immune function and stimulate a localized hyperinflammatory response [13]. Although *sn*-2 palmitic TAGs have shown some immunity benefits on the host, the influence of its major TAG composition on immunity function, and related inflammatory responses and oxidative stress is still lacking.

In this study, nontargeted metabolic and lipidomic profiling analysis by using ultra-high performance liquid chromatography coupled with mass spectrometry (UHPLC-MS) was applied to explore the metabolic alterations of rats, which were fed with control fat or HMFS with different total *sn*-2 palmitic TAGs and OPL to OPO ratios. Meanwhile, the targeted biochemical factors were further analyzed to unveil the influence of HMFS on the inflammatory responses, immunity function, and oxidative stress in rats. The study may help to better understand the metabolic regulatory effects of the structure and composition of TAGs in HMFS on the host.

## 2. Materials and Methods

### 2.1. Chemicals

Enzyme-linked immunosorbent assay (ELISA) kits for interleukin-6 (IL-6), interleukin-4 (IL-4), tumor necrosis factor-α (TNF-α), immunoglobulin A (IgA), immunoglobulin M (IgM), superoxide dismutase (SOD), glutathione peroxidase (GSH-px), and malondialdehyde (MDA) were all obtained from Dakewe Biotech Co., Ltd. (Shanghai, China). LC-MS (liquid chromatography-mass spectrometry) grade methanol, acetonitrile, and formic acid were purchased from Thermo Fisher Scientific (Waltham, MA, USA). Chlorophenylalanine was purchased from Anpel Laboratory Technologies, Inc. (Shanghai, China). All other chemical reagents were of the highest quality and purchased from Sigma-Aldrich (St. Louis, MO, USA).

### 2.2. Experimental Animals and Treatment

Since this study is a continuation of our previous work, the animals were fed and grouped as reported in our recently published literature [8]. Briefly, forty male SD rats were randomly selected into four groups (*n* = 10 rats per group) after one week acclimatized feeding, which were then fed with control fat (CF) and 3 kinds of HMFS for 4 weeks, respectively. The CF and HMFS were kindly provided by Kerry Oils & Grains Industries Co., Ltd. (Shanghai, China), and the detailed chemical composition of CF (*sn*-2 palmitic acid level of 15.54%, OPL to OPO ratio of 0.4), HMFS1 (*sn*-2 palmitic acid level of 54.36%, OPL to OPO ratio of 0.3), HMFS2 (*sn*-2 palmitic acid level of 60.02%, OPL to OPO ratio of 0.9), and HMFS3 (*sn*-2 palmitic acid level of 57.87%, OPL to OPO ratio of 1.4) is shown in Appendix A. The four experimental diets were made from 15 wt% CF or HMFS mixed with commercial non-fat diets, and the detailed diet components are shown in Appendix A. All animal procedures were conducted in strict accordance with the laboratory animal guidelines, which were approved by the Animal Care and Use Committee of Shanghai Jiao Tong University (A2020078).

### 2.3. Serum Collection

After the 4-week experimental feeding, the SD rats were anesthetized with CO_2_ and whole blood was collected from the abdominal aorta. Serum was obtained from the blood samples by centrifugation (1300× *g*, 10 min, 4 °C), and then stored at −80 °C until analysis [14].

### 2.4. Serum Metabolomics Analysis

Serum metabolomics was analyzed according to a previously reported literature with some modifications [15]. Rat serum (50 μL) was mixed with 200 μL of extract solvent (acetonitrile: methanol = 1:1, containing 2 μg/mL of chlorophenylalanine) and vortexed for 30 s. After resting at −20 °C for at least 2 h, 150 μL of supernatant was obtained after centrifugation (13,400× *g*, 20 min, 4 °C). The supernatant was concentrated to dryness using a centrifugal concentrator. A total of 50 μL of solvent (methanol:water = 3:7) was added for redissolving and then vortexed mixing. After centrifugation, the supernatant was taken in the injection vial for measurement.

An UHPLC system (Vanquish, Thermo Fisher Scientific, Bremen, Germany) coupled with a Q exactive plus mass spectrometer (Orbitrap MS) was carried out for the analysis of metabolic profiles at both ESI positive (+) and negative (−) ion modes. Chromatographic separation was performed on a Waters ACQUITY UPLC HSS T3 column (100 mm × 2.1 mm, 1.7 μm). The column temperature was 40 °C, the flow rate was 0.4 mL/min, and the injection volume was 1.0 μL. The gradient mobile phase comprised of 0.1% formic acid aqueous solution (A) and 0.1% formic acid acetonitrile solution (B). The gradient program was as follows: 0–12 min, 1–100% B; 12–13 min, held at 100% B. The conditions of the mass spectrometer were as follows: scan range, 67–1000 amu; spray voltage, 3.2 kV (positive mode) and 2.8 kV (negative mode); capillary temperature, 320 °C. Raw chromatographic and MS data of serum metabolites were acquired by Xcalibur 3.0, and processed using Progenesis QI v2.3 (Waters Co., Milford, MA, USA) for peak picking, alignment, and normalization. UNIFI software version 1.8 and an online database [human metabolome database (HMDB)] were used to identify the metabolites in four experimental groups.

### 2.5. Serum Lipidomics Analysis

Serum lipidomics analysis was carried out following a literature published before with some modifications [16]. Serum (50 μL) was mixed with 200 μL of methanol and vortexed sufficiently. A total of 400 μL of trichloromethane was added to the mixed solution, vortexed thoroughly, and shaken for 1 h at room temperature. A total of 170 μL of ultrapure water was added to the solution and vortexed thoroughly. After resting at 4 °C for 10 min, the bottom trichloromethane layer was obtained after centrifugation (13,400× *g*, 10 min, 4 °C). The supernatant was added to 400 μL of washing solution (trichloromethane: methanol: water = 85:14:1), and the two lower layers collected were combined after centrifugation. A total of 100 μL of solvent (dichloromethane: isopropanol: methanol = 1:1:2) was added for redissolving and then vortexed for 1 min. After centrifugation, the supernatant was taken in the injection vial for measurement.

A UHPLC system (Vanquish, Thermo Fisher Scientific) coupled with a Q exactive plus mass spectrometer (Orbitrap MS, Thermo) was used for the analysis of lipids at both ESI positive (+) and negative (−) ion modes. Waters ACQUITY UPLC BEH C18 (100 mm × 2.1 mm, 1.7 μm) was used for the chromatographic separation. The column temperature was 55 °C, the flow rate was 0.4 mL/min, and the injection volume was 1.0 μL. The gradient mobile phase comprised acetonitrile/water (60:40) with 10 mM ammonium formate and 0.1% formic acid (A), and isopropanol/acetonitrile (90:10) with 10 mM ammonium formate and 0.1% formic acid (B). The gradient program was as follows: 0–17 min, 5–100% B. The conditions of the mass spectrometer were as follows: scan range, 150–2000 amu; spray voltage, 3.8 kV (positive mode) and 3.0 kV (negative mode); capillary temperature, 320 °C. Raw chromatographic and MS data of serum lipids were acquired by Xcalibur 3.0, and processed by Lipidsearch 4.2 to obtain lipid information.

### 2.6. Serum Immunity-Related Parameter and Antioxidant Parameter Measurement

Serum immunity-related parameters and antioxidant parameters, including IL-6, IL-4, TNF-α, IgA, IgM, SOD, GSH-px, and MDA, were determined using ELISA kits according to the manufacturer’s instructions, respectively.

### 2.7. Statistical Analysis

Principal component analysis (PCA), volcano plot analysis, and pathway analysis were performed using MetaboAnalyst 6.0 software. Statistics were analyzed using SPSS 24.0 and statistical significance was calculated by one-way ANOVA and Tukey’s post hoc multiple comparisons (*p* < 0.05 or 0.01). All the figures were made by using GraphPad Prism 8.0. The correlation analysis was performed using Spearman’s correlation analysis method.

## 3. Results and Discussion

### 3.1. Metabolomics and Lipidomics Profiling

PCA is a linear dimensionality reduction technique applied to examine the overall profile among samples and the consistency of the analytical process [17]. Therefore, PCA was first used to detect serum metabolomics and lipidomics profiles of CF or HMFS-fed rats. As shown in Figure 1, a clear separation in the PCA score plots was observed among the CF and HMFS-fed rat groups both at the positive ion mode and negative ion mode, no matter for metabolomics (Figure 1A,B) or lipidomics profiles (Figure 1C,D), which indicated that the total *sn*-2 palmitic TAGs and OPL to OPO ratio in HMFS had some influence on the serum metabolites and lipids in rats.

### 3.2. Metabolomics Analysis

The identified serum metabolites between different experimental groups were utilized to generate the volcano plots (Appendix A). These significantly altered serum metabolites were further screened and identified as differential metabolites based on the fold change (FC) > 2 or <0.5, and variable importance in projection (VIP) > 1. Finally, a total of 16 serum differential metabolites were obtained among four experimental groups, including 3 glycerophospholipids, 1 bile acid, 2 organosulfonic acids, 5 indoles and derivatives, 3 amino acids and derivatives, 2 purines and purine derivatives (Table 1). These differential metabolites were further used for the metabolic pathway analysis. Based on the *p* < 0.05 and rich factor > 0.1, the differential metabolic pathways were further filtered out.

Compared to the CF-fed rat group, increasing the *sn*-2 palmitic acid content from 15.54% to 54.36% in HMFS1 could significantly up-regulate the abundances of taurocholic acid, 6-hydroxymelatonin, 5-hydroxy-L-tryptophan, L-tryptophan, L-glutamine, and creatine, and down-regulate the abundances of glycerol 3-phosphate, lysophosphatidylcholine (lysoPC) (18:0), and xanthosine (*p* < 0.05 or 0.01) (Table 1). Meanwhile, a total of four differential metabolic pathways, including tryptophan metabolism, glycerophospholipid metabolism, purine metabolism, and bile acid biosynthesis were found between CF and HMFS1-fed rats (Figure 2A). When the *sn*-2 palmitic acid content of HMFS was maintained in the range of 54.36% to 60.02%, increasing the OPL to OPO ratio in HMFS from 0.3 (HMFS1) to 0.9 (HMFS2) could further increase the abundance of L-tryptophan and decrease the abundance of xanthosine (*p* < 0.01). Meanwhile, other four differential metabolites were also observed in HMFS2-fed rats, including taurine, glycine, L-cysteine, and hypoxanthine. As shown in Figure 2B, three differential metabolic pathways of taurine and hypotaurine metabolism, glycine, serine and threonine metabolism, and bile acid biosynthesis were identified between HMFS1 and HMFS2 groups. When the OPL to OPO ratio in HMFS further increased to 1.4 (HMFS3), a new differential metabolic pathway of cysteine and methionine metabolism was found between the HMFS1 and HMFS3-fed rats (*p* < 0.01), besides six differential metabolic pathways mentioned earlier (Figure 2C). For the metabolites involved in the cysteine and methionine metabolism pathway, L-cysteine was significantly increased (*p* < 0.01), indicating the up-regulated cysteine and methionine metabolism pathway in HMFS3-fed rats.

In our recent study, compared to CF-fed rats, the increased levels of total serum bile acids, especially for the tauro-conjugated bile acids were observed in HMFS-fed rats [8], which might be explained by the up-regulated bile acid biosynthesis pathway observed in this study. Meanwhile, glycerophospholipid metabolism was one of the essential lipid metabolism pathways, which showed a strongly positive association with hyperlipidemia and obesity in mice [18,19]. Qu et al. found that the tripeptide DT-109 (Gly-Gly-Leu) dose-dependently attenuated hepatic steatosis in mice through up-regulating glycine, serine and threonine metabolism [20]. In this study, it was found that increasing total *sn*-2 palmitic TAGs and OPL to OPO ratio in HMFS could down-regulate the glycerophospholipid metabolism and meanwhile up-regulate glycine, serine and threonine metabolism pathway. These results may help us better understand our previous observations that HMFS-fed rats had reduced body weight and accumulation of lipid droplets in liver and perirenal adipose tissue, and improved serum lipid indicators [8]. The taurine and hypotaurine metabolism is an energy metabolic homeostasis-related signaling pathway. According to the observations of Zhu et al., *Brassica rapa* L. extract could strengthen energy metabolism in mice and help against fatigue through up-regulating taurine and hypotaurine metabolism pathways [21]. In this study, increasing the OPL to OPO ratio in HMFS also significantly up-regulated the taurine and hypotaurine metabolism in a dose-dependent manner, which provides further evidence for the significantly higher expressions of two key thermogenic proteins (peroxisome proliferators-activated receptor γ coactivator lalpha and uncoupling protein 1) in perirenal adipose tissue of HMFS-fed rats, especially for the rats fed with HMFS3 (OPL to OPO ratio of 1.4) [22].

Furthermore, a recent study has shown that cysteine and methionine metabolism, tryptophan metabolism, and purine metabolism all have a strong correlation with the inflammatory responses and immunity function in COVID-19 patients [23]. In addition, Zhao et al. found that the up-regulation of glycerophospholipid metabolism might exacerbate oxidative damage and induce hyperinflammatory response in zebrafish [24]. Therefore, serum biochemical indicators related to the inflammatory response, oxidative stress, and immunity function were also measured in this study. As shown in Figure 3, the statistically lower pro-inflammatory factor levels, including IL-6 and TNF-α, were observed for HMFS-fed rats, compared to that of CF-fed rats. The lowest levels of IL-6 and TNF-α were also shown in HMFS3-fed rats (168.32 pg/mL and 244.72 pg/mL) (Figure 3A,B). Meanwhile, the level of anti-inflammatory factor IL-4 was just the opposite, and the highest IL-4 was shown in HMFS3-fed rats (79.34 pg/mL) (Figure 3C). Moreover, the correlation between these inflammatory factors and serum differential metabolites was analyzed. As shown in Figure 4A, IL-6 level was observed to be negatively correlated with the L-tryptophan and L-cysteine, while positively correlated with hypoxanthine. Moreover, there was a significant negative correlation between TNF-α and taurine or L-tryptophan, and a significant positive correlation between TNF-α and hypoxanthine. Taurine, L-tryptophan, and hypoxanthine were crucial metabolites involved in taurine and hypotaurine metabolism, tryptophan metabolism, and purine metabolism. L-cysteine was involved in cysteine and methionine metabolism. These observations could further indicate that the increased *sn*-2 palmitic TAGs and OPL to OPO ratio in HMFS might reduce the inflammatory response in rats by up-regulating taurine and hypotaurine metabolism, tryptophan metabolism, and cysteine and methionine metabolism, and down-regulating purine metabolism. A previous study also found that mogroside V could reduce the TNF-α level in serum and attenuate lung inflammation through the up-regulation of taurine and hypotaurine metabolism in asthmatic mice [25]. Xiao et al. found that the decreased IL-6 level in serum and up-regulated cysteine and methionine metabolism were shown in healthy people, in comparison to COVID-19 patients [23].

Immunoglobulin, including IgA and IgM, is a kind of glycosylated protein, which plays a critical role in defending against a variety of pathogenic infections [26]. As shown in Figure 3D,E, compared to that of CF-fed rats, three groups of HMFS-fed rats all exhibited a significant increase in the IgA level (*p* < 0.05), with the highest IgA level shown in the HMFS3-fed rats (300.70 μg/mL). Meanwhile, there is no significant difference in the IgM level among the four experimental rat groups. Based on the Spearman correlation analysis, the IgA level was positively correlated with L-tryptophan involved in the tryptophan metabolism pathway. Shi et al. showed consistent observations that the activation of IgA could be modulated by up-regulating the tryptophan metabolism pathway in high-fat-diet mice [27]. Oxidative stress induces cellular damage, which is often characterized by decreased levels of antioxidant enzymes, and increased levels of MDA [28]. As shown in Figure 3F,G, the increased levels of antioxidant enzymes, including SOD and GSH-Px were both observed in three HMFS-fed rat groups, compared with those of the CF-fed rat group. The highest SOD and GSH-Px levels were shown in HMFS3-fed rats (226.94 U/mL and 643.87 U/mL). Moreover, a statistically lower MDA level was observed for HMFS1-fed rats (5.03 nmol/mL) compared to that of CF-fed rats (6.20 nmol/mL). The MDA level of HMFS3-fed rats was further significantly decreased to 3.25 nmol/mL, compared to those of the other two HMFS-fed rat groups (*p* < 0.05) (Figure 3H). Meanwhile, the correlation between serum antioxidant parameters and differential metabolites was also analyzed (Figure 4A). MDA was observed to be positively correlated with the lysoPC (18:0) and hypoxanthine, which were crucial metabolites involved in glycerophospholipid metabolism and purine metabolism, respectively (*p* < 0.05). Meanwhile, the significantly negative correlations between SOD or GSH-Px and lysoPC (18:0) or hypoxanthine in rats could also be found. Han et al. have shown that the hydroxytyrosol treatment could increase the activities of SOD and GSH-Px, and decrease the MDA level in mice serum through down-regulating the glycerophospholipid metabolism pathway [29]. Meanwhile, a previous study showed that the chrysanthemum extract could increase SOD and GSH-Px activities in LO2 hepatocytes by down-regulating the purine metabolism pathway [30].

### 3.3. Lipidomics Analysis

By using the lipidomics technique, the serum lipids of CF and HMFS-fed rats were identified, which were first utilized to generate the volcano plots (Appendix A). According to FC > 2 or <0.5, *p* < 0.05, and VIP > 1, a total of 4 differential lipid species, including fatty acids, glycerophospholipids, glycerolipids, and sphingolipids, were further identified (Table 2). Specifically, the significantly down-regulated fatty acid (FA) (22:5), lysoPC (18:0), lysoPC (18:2), and lysoPC (20:1), lysophosphatidyl ethanolamine (lysoPE) (18:0), lysoPE (18:2), lysoPE (20:4), lysoPE (22:6), phosphatidylcholine (PC) (16:0_16:0), PC (18:1_22:6), triacylglycerol (TG) (16:0_16:0_16:0), TG (16:0_16:0_18:1), and sphingomyelin (SM) (d18:1_16:0) were observed in HMFS1-fed rats compared to CF-fed rats. Based on the metabolic pathway analysis (Figure 2D), these differential lipids were mainly involved in glycerophospholipid metabolism and glycerolipid metabolism, suggesting that increasing the total *sn*-2 palmitic TAGs could down-regulate glycerophospholipid metabolism and glycerolipid metabolism in rats. In addition, increasing the OPL to OPO ratio in HMFS from 0.3 (HMFS1) to 0.9 (HMFS2) could further decrease the abundance of lysoPE (18:2) and TG (16:0_16:0_16:0) (*p* < 0.01). Meanwhile, other three differential lipids were also decreased in HMFS2-fed rats, that was lysoPC (20:0), PE (18:0_18:2), and PE (18:1_18:1). These differential serum lipids were mainly involved in the glycerophospholipid metabolism pathway (Figure 2E). Moreover, increasing the OPL to OPO ratio from 0.3 (HMFS1) to 1.4 (HMFS3), all the above-mentioned differential lipids were further significantly down-regulated, except for lysoPC (20:1) and SM (d18:1_16:0). Meanwhile, an additional metabolic pathway of sphingolipid metabolism was observed between HMFS1 and HMFS3-fed rats (Figure 2F). It is known that glycerophospholipid metabolism, glycerolipid metabolism, and sphingolipid metabolism are all related to the lipid metabolism homeostasis in the host. Wei et al. found that fermented tomatoes could reduce serum total cholesterol (TC) and triacylglycerol (TAG) levels and regulate lipid metabolism in high-fat diet-induced mice through down-regulating glycerophospholipid metabolism and sphingolipid metabolism [31]. Meanwhile, Feng et al. observed that tangeretin could decrease serum TC and low density lipoprotein-cholesterol (LDL-c) by down-regulating glycerophospholipid metabolism and glycerolipid metabolism in high-fat diet-fed rats [32].

To further clarify the relationship between these differential lipids (Table 2) and serum lipid indicators (TC, TAG, HDL-c, and LDL-c) measured in our previous study [8], the association analysis was also performed (Figure 4B). As shown in Figure 4B, significant positive correlations between serum lysoPCs and TC or LDL-c were observed. A previous study correlated well with our observations, which also showed that there was a significant positive correlation between the TC level and lysoPCs in the serum of hypercholesterolemia patients, possibly due to the fact that lysoPC could promote cholesterol-related small-molecular lipids and lipoprotein accumulation [33]. Meanwhile, lysoPC is also known as the hydrolyzed product of LDL-c, so it is reasonable that the significantly positive correlations between serum lysoPCs and LDL-c are observed [34]. Moreover, there were obvious positive correlations between serum TG species and TC or TAG, and negative correlations between serum TG species and HDL-c. These correlation results were also consistent with some previously published studies that serum TG species had a strong positive correlation with TC or TAG, and a negative correlation with HDL-c [35,36].

## 4. Conclusions

In conclusion, an integrated strategy combining metabolomics and lipidomics was used to explore the influence of *sn*-2 palmitic TAGs and OPL to OPO ratio on metabolic alternation in rats. The metabolic pathway showed that increasing the *sn*-2 palmitic TAGs and ratio of OPL to OPO in HMFS could significantly up-regulate glycine, serine and threonine metabolism, bile acid biosynthesis and taurine and hypotaurine metabolism, and down-regulate glycerophospholipid metabolism, glycerolipid metabolism and sphingolipid metabolism, which are mainly involved in lipid, bile acid and energy metabolism in rats. Meanwhile, both the metabolic profiling analysis and biochemical factors measurement suggested that increasing the *sn*-2 palmitic TAGs and OPL to OPO ratio in HMFS could reduce the inflammatory response and oxidative stress, and improve the immunity function of rats mainly by regulating glycerophospholipid metabolism, purine metabolism, tryptophan metabolism, cysteine and methionine metabolism, and taurine and hypotaurine metabolism. These results will help to improve our understanding of the regulatory effect of TAG structure and composition in HMFS on the nutritional functions of the host, which could further allow us to design infant formula for improving the overall health of infants.

## Figures and Tables

**Figure 1 nutrients-16-01299-f001:**
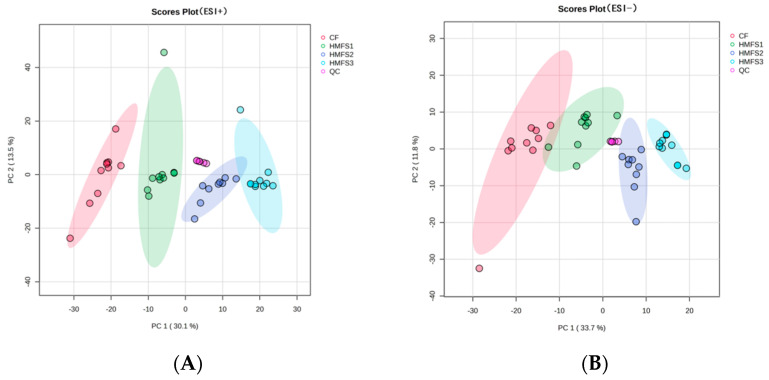
Serum principal components analysis (PCA) score plots for Sprague–Dawley rats. Positive ion (**A**) and negative ion (**B**) mode in metabolomics, positive ion (**C**) and negative ion (**D**) mode in lipidomics. CF, control fat group; HMFS1, human milk fat substitute 1 group; HMFS2, human milk fat substitute 2 group; HMFS3, human milk fat substitute 3 group.

**Figure 2 nutrients-16-01299-f002:**
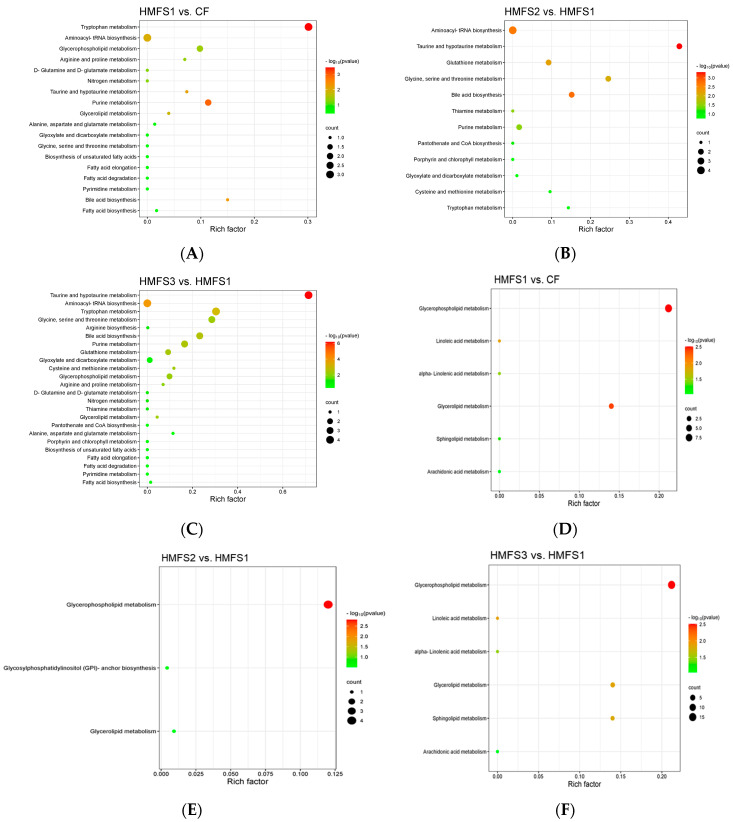
Metabolic pathway analysis of serum differential metabolites (**A**–**C**) and lipids (**D**–**F**) in Sprague–Dawley rats. The color of dots indicated the *p*-value level, and the size of dots indicated the count of differential metabolites involved in a certain metabolic pathway. CF, control fat group; HMFS1, human milk fat substitute 1 group; HMFS2, human milk fat substitute 2 group; HMFS3, human milk fat substitute 3 group.

**Figure 3 nutrients-16-01299-f003:**
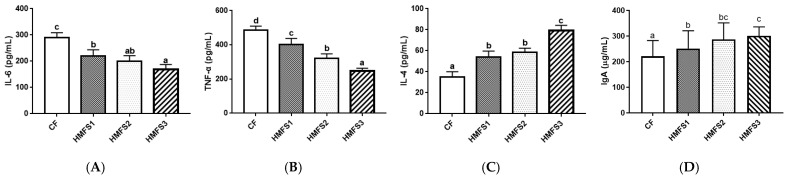
Serum immunity-related parameters for IL-6 (**A**), TNF-α (**B**), IL-4 (**C**), IgA (**D**), and IgM (**E**) in Sprague–Dawley rats. Serum antioxidant parameters for SOD (**F**), GSH-px (**G**), and MDA (**H**) in Sprague–Dawley rats. The vertical bars stand for the standard deviation of data (*n* = 10). Different letters stand for significant differences among four dietary fat formula groups (*p* < 0.05). IL-6, interleukin-6; TNF-α, tumor necrosis factor-α; IL-4, interleukin-4, IgA, immunoglobulin A; IgM, immunoglobulin M; SOD, superoxide dismutase; GSH-px, glutathione peroxidase; MDA, malondialdehyde. CF, control fat group; HMFS1, human milk fat substitute 1 group; HMFS2, human milk fat substitute 2 group; HMFS3, human milk fat substitute 3 group.

**Figure 4 nutrients-16-01299-f004:**
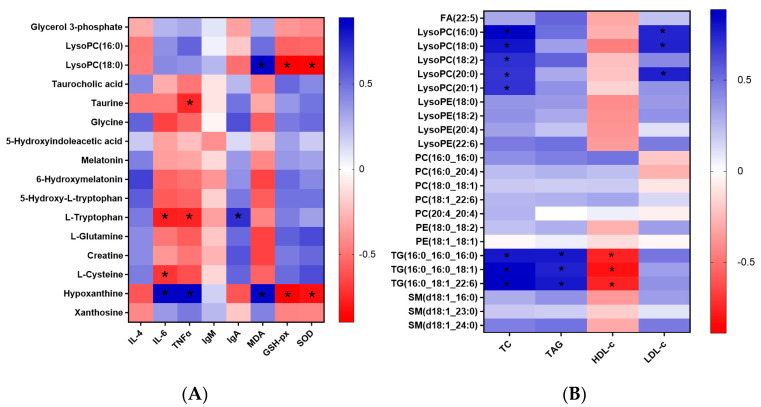
Heatmap of Spearman correlation analysis between serum differential metabolites and serum immunity-related parameters or antioxidant parameters measurement (**A**). Heatmap of Spearman correlation analysis between serum differential lipids and serum lipid parameters (TC, TAG, HDL-c, and LDL-c). The serum lipid parameters have been published in our previous article [8] (* *p* < 0.05) (**B**). IL-6, interleukin-6; TNF-α, tumor necrosis factor-α; IL-4, interleukin-4, IgA, immunoglobulin A; IgM, immunoglobulin M; SOD, superoxide dismutase; GSH-px, glutathione peroxidase; MDA, malondialdehyde; LysoPC(16:0), lysophosphatidyl choline(16:0); LysoPC(18:0), lysophosphatidyl choline(18:0); FA, fatty acid; LysoPC, lysophosphatidyl choline; LysoPE, lysophosphatidyl ethanolamine; PC, phosphatidyl choline; PE, phosphatidyl ethanolamine; TG, triacylglycerol; SM, sphingomyelin.

**Table 1 nutrients-16-01299-t001:** Identification of serum differential metabolites in Sprague–Dawley rats.

Class	Ion Mode	Identity	HMDB ID	Observed *m*/*z*	RT (min)	Formula	Mass Error (ppm)	HMFS1 vs. CF			HMFS2 vs. HMFS1			HMFS3 vs. HMFS1		
								VIP	FC	Trend	VIP	FC	Trend	VIP	FC	Trend
Glycerophospholipids	−	Glycerol 3-phosphate	HMDB00126	171.0065	0.65	C_3_H_9_O_6_P	0.31	1.23	0.37	↓ **	0.91	0.97	↓	2.88	0.21	↓ **
	+	LysoPC (16:0)	HMDB10382	478.3291	7.50	C_24_H_50_NO_7_P	−0.17	1.13	0.82	↓	0.88	0.92	↓	1.54	0.05	↓ **
	+	LysoPC (18:0)	HMDB10384	524.3713	6.06	C_26_H_54_NO_7_P	0.53	1.24	0.45	↓ **	0.65	0.93	↓	1.83	0.33	↓ **
Bile acids	+	Taurocholic acid	HMDB00036	538.2805	5.07	C_26_H_45_NO_7_S	−0.18	1.01	2.06	↑ *	1.65	1.18	↑	1.91	3.08	↑ **
Organosulfonic acids	+	Taurine	HMDB00251	126.0219	0.62	C_2_H_7_NO_3_S	−0.24	1.18	1.29	↑	1.48	1.02	↑	2.11	2.94	↑ **
	−	Taurine	HMDB00251	124.0073	0.61	C_2_H_7_NO_3_S	0.03	1.33	1.11	↑	1.76	1.01	↑	1.98	3.16	↑ **
	−	Glycine	HMDB00123	74.0246	0.65	C_2_H_5_NO_2_	−1.41	1.09	1.10	↑	1.84	5.57	↑ **	2.43	6.02	↑ **
Indoles and derivatives	+	5-Hydroxyindoleacetic acid	HMDB00763	192.0656	3.13	C_10_H_9_NO_3_	0.28	1.09	0.70	↓	0.93	1.63	↑	1.98	3.44	↑ **
	−	5-Hydroxyindoleacetic acid	HMDB00763	190.0510	3.15	C_10_H_9_NO_3_	0.16	1.18	0.86	↓	0.88	1.33	↑	1.69	3.01	↑ **
	+	Melatonin	HMDB01389	255.1056	4.76	C_13_H_16_N_2_O_2_	−0.63	1.52	2.45	↑	1.63	1.51	↑	1.21	8.36	↑ **
	−	6-Hydroxymelatonin	HMDB04081	293.1142	4.09	C_13_H_16_N_2_O_3_	−0.19	1.98	2.79	↑ *	0.82	1.06	↑	1.83	2.48	↑ **
	+	5-Hydroxy-L-tryptophan	HMDB00472	221.0920	2.74	C_11_H_12_N_2_O_3_	0.19	1.09	7.61	↑ **	1.02	1.49	↑	0.44	1.08	↑
	−	5-Hydroxy-L-tryptophan	HMDB00472	219.0774	2.74	C_11_H_12_N_2_O_3_	−0.24	1.16	8.13	↑ **	1.29	1.27	↑	0.72	1.41	↑
	+	L-Tryptophan	HMDB00929	205.0971	3.87	C_11_H_12_N_2_O_2_	−0.10	1.63	2.04	↑ **	1.26	2.54	↑ **	1.29	3.11	↑ **
	−	L-Tryptophan	HMDB00929	203.0826	3.87	C_11_H_12_N_2_O_2_	0.33	1.82	2.21	↑ **	1.04	2.15	↑ **	1.43	3.28	↑ **
Amino acids and derivatives	−	L-Glutamine	HMDB00641	145.0619	0.63	C_5_H_10_N_2_O_3_	0.52	1.69	3.46	↑ **	1.77	1.00	—	2.05	3.15	↑ **
	−	Creatine	HMDB00064	130.0622	0.73	C_4_H_9_N_3_O_2_	0.23	1.25	7.06	↑ **	0.88	1.10	↑	1.67	3.10	↑ **
	−	L-Cysteine	HMDB00574	120.0125	1.00	C_3_H_7_NO_2_S	0.11	0.77	0.98	↓	2.00	4.25	↑ **	1.32	4.28	↑ **
Purines and purine derivatives	+	Hypoxanthine	HMDB00157	137.0458	3.54	C_5_H_4_N_4_O	0.55	0.80	0.60	↓	1.87	0.37	↓ **	1.74	0.18	↓ **
	+	Xanthosine	HMDB00299	285.0826	3.31	C_10_H_12_N_4_O_6_	−1.42	2.71	0.22	↓ **	1.76	0.22	↓ **	1.82	0.11	↓ **
	−	Xanthosine	HMDB00299	283.0685	3.34	C_10_H_12_N_4_O_6_	−0.21	2.55	0.39	↓ **	1.98	0.37	↓ **	1.95	0.06	↓ **

* and ** stand for *p* < 0.05 and *p* < 0.01, respectively. ↑ and ↓ stand for the increased or decreased trend of certain metabolite, respectively. LysoPC (16:0), lysophosphatidyl choline (16:0); LysoPC (18:0), lysophosphatidyl choline (18:0). CF, control fat group; HMFS1, human milk fat substitute 1 group; HMFS2, human milk fat substitute 2 group; HMFS3, human milk fat substitute 3 group.

**Table 2 nutrients-16-01299-t002:** Identification of serum differential lipids in Sprague–Dawley rats.

Class	Ion Mode	Identity	HMDB ID	Observed m/z	RT (min)	Formula	Mass Error (ppm)	HMFS1 vs. CF			HMFS2 vs. HMFS1			HMFS3 vs. HMFS1		
								VIP	FC	Trend	VIP	FC	Trend	VIP	FC	Trend
Fatty acids	−	FA (22:5)	HMDB0247288	329.2486	3.94	C_22_H_34_O_2_	0.01	1.84	0.42	↓ **	0.92	0.65	↓	1.63	0.43	↓ **
Glycerophospholipids	+	LysoPC (16:0)	HMDB0010382	496.3398	2.15	C_24_H_50_O_7_NP	1.20	0.76	0.97	↓	0.85	0.93	↓	1.52	0.45	↓ **
	+	LysoPC (18:0)	HMDB0010384	524.3711	2.40	C_26_H_54_O_7_NP	0.75	1.59	0.45	↓ **	0.92	0.98	↓	1.81	0.31	↓ **
	+	LysoPC (18:2)	HMDB0010386	520.3398	2.22	C_26_H_50_O_7_NP	0.37	1.51	0.35	↓ **	0.82	0.87	↓	1.15	0.45	↓ **
	+	LysoPC (20:0)	HMDB0010390	552.4024	3.96	C_28_H_58_O_7_NP	0.68	0.91	1.00	—	1.14	0.38	↓ **	1.36	0.31	↓ **
	+	LysoPC (20:1)	HMDB0010391	550.3867	3.36	C_28_H_56_O_7_NP	0.35	1.91	0.31	↓ **	0.75	0.82	↓	0.78	0.65	↓
	−	LysoPE (18:0)	HMDB0011130	480.3096	3.67	C_23_H_48_O_7_NP	0.69	1.63	0.49	↓ **	0.87	1.00	—	1.03	0.45	↓ **
	−	LysoPE (18:2)	HMDB0011507	476.2783	2.16	C_23_H_44_O_7_NP	0.47	1.81	0.38	↓ **	1.75	0.44	↓ **	1.38	0.22	↓ **
	−	LysoPE (20:4)	HMDB0011487	500.2783	2.14	C_25_H_44_O_7_NP	1.32	1.59	0.33	↓ **	0.95	0.88	↓	1.55	0.41	↓ **
	−	LysoPE (22:6)	HMDB0011496	524.2783	2.28	C_27_H_44_O_7_NP	0.25	1.21	0.28	↓ **	0.74	0.97	↓	1.98	0.34	↓ **
	+	PC (16:0_16:0)	HMDB0000564	756.5514	7.87	C_40_H_80_O_8_NP	0.01	1.24	0.40	↓ **	0.85	0.82	↓	1.62	0.49	↓ **
	+	PC (16:0_20:4)	HMDB0007982	804.5514	8.41	C_44_H_80_O_8_NP	1.05	0.76	0.75	↓	0.98	0.94	↓	1.54	0.42	↓ **
	+	PC (18:0_18:1)	HMDB0008037	810.5983	10.97	C_44_H_86_O_8_NP	1.41	0.91	0.90	↓	0.94	0.93	↓	1.54	0.40	↓ **
	+	PC (18:1_22:6)	HMDB0008090	854.5670	7.96	C_48_H_82_O_8_NP	0.01	1.24	0.48	↓ **	0.56	0.74	↓	1.39	0.40	↓ **
	+	PC (20:4_20:4)	HMDB0008444	852.5514	6.72	C_48_H_80_O_8_NP	0.01	0.52	0.96	↓	0.73	0.64	↓	1.98	0.45	↓ **
	−	PE (18:0_18:2)	HMDB0008994	742.5392	10.72	C_41_H_78_O_8_NP	1.29	0.32	0.97	↓	1.62	0.32	↓ **	1.32	0.31	↓ **
	−	PE (18:1_18:1)	HMDB0009025	742.5392	7.33	C_41_H_78_O_8_NP	0.02	0.68	0.88	↓	1.61	0.45	↓ **	1.28	0.23	↓ **
Glycerolipids	+	TG (16:0_16:0_16:0)	HMDB0005356	829.7256	12.25	C_51_H_98_O_6_	1.05	1.59	0.49	↓ **	1.52	0.07	↓ **	1.67	0.05	↓ **
	+	TG (16:0_16:0_18:1)	HMDB0005360	855.7412	14.75	C_53_H_100_O_6_	1.51	1.38	0.44	↓ **	0.97	0.74	↓	1.18	0.24	↓ **
	+	TG (16:0_18:1_22:6)	HMDB0044135	905.7593	12.24	C_59_H_100_O_6_	0.17	0.94	0.81	↓	0.88	0.76	↓	1.28	0.36	↓ **
Sphingolipids	+	SM (d18:1_16:0)	HMDB0010169	725.5568	8.01	C_39_H_79_O_6_N_2_P	1.48	1.11	0.45	↓ **	0.98	0.83	↓	0.42	0.74	↓
	+	SM (d18:1_23:0)	HMDB0012105	823.6663	11.65	C_46_H_93_O_6_N_2_P	0.71	0.53	0.77	↓	0.93	0.82	↓	1.73	0.35	↓ **
	+	SM (d18:1_24:0)	HMDB0011697	837.6820	11.83	C_47_H_95_O_6_N_2_P	1.50	0.96	0.88	↓	0.89	0.72	↓	1.92	0.48	↓ **

** Stands for *p* < 0.01. ↓ stands for the decreased trend of certain metabolite. FA, fatty acid; LysoPC, lysophosphatidyl choline; LysoPE, lysophosphatidyl ethanolamine; PC, phosphatidyl choline; PE, phosphatidyl ethanolamine; TG, triacylglycerol; SM, sphingomyelin. CF, control fat group; HMFS1, human milk fat substitute 1 group; HMFS2, human milk fat substitute 2 group; HMFS3, human milk fat substitute 3 group.

## Data Availability

The data that support the findings of this study are available from the corresponding author upon reasonable request.

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
