# Peer review of "Effects of sn-2 Palmitic Triacylglycerols and the Ratio of OPL to OPO in Human Milk Fat Substitute on Metabolic Regulation in Sprague-Dawley Rats"

_nutrients, 2024, doi:10.3390/nu16091299_

Round 1

Reviewer 1 Report

Comments and Suggestions for Authors

Journal: MDPI_nutrients 

Title: Effects of Sn-2 Palmitic Triacylglycerols and the Ratio of OPL to OPO in Human Milk Fat Substitute on Metabolic Regulation in Sprague-Dawley Rats 

Manuscript ID: nutrients_2951049

 Comments 

This study investigated the “Effects of Sn-2 Palmitic Triacylglycerols and the Ratio of OPL to OPO in Human Milk Fat Substitute on Metabolic Regulation in Sprague-Dawley Rats”. Metabolomics and lipidomics were used to explore the influence of sn-2 palmitic TAGs and OPL to OPO ratio on metabolic alternation in rats. These results will help to improve our understanding of the regulatary effect of TAG structure and composition in HMFS on the nutritional functions on host, which could further allow us to design infant formula for improving the overall health of infants.

This study contains some interesting research results. And, this is a meaningful study in that it helps to undetstand the nutritional functions of TAG structure and composition in infant formula. But, in order to submit this manuscript to MDPI nutrients, the following comments need to be revised.

1.      Please suggest how you verified the analytical accuracy and precision of the Q exactive plus MS instrument for metabolomics and lipidomics of serum collection samples.

2.      Please provide the detailed analysis information and related references in Experimental Methods.

3.      Is the population size adequate to explain the results of this study?

Overall, please double-check the spacing (units, spelling, and etc) and the writing references according to the guidelines of the Journal.

Author Response

Manuscript ID: nutrients-2951049

Title: Effects of Sn-2 Palmitic Triacylglycerols and the Ratio of OPL to OPO in Human Milk Fat Substitute on Metabolic Regulation in Sprague-Dawley Rats

Authors: Lin Zhu, Shuaizhen Fang, Yaqiong Zhang*, Xiangjun Sun, Puyu Yang, Weiying Lu, Liangli (Lucy) Yu

1. Reviewer #1:Please suggest how you verified the analytical accuracy and precision of the Q exactive plus MS instrument for metabolomics and lipidomics of serum collection samples. 

Response: Done. Firstly, the quality control samples were all clustered in the center of PCA score plots, no matter for the metabolomics or lipidomics results, which suggested that the analytical method was accurate (Figure 1). Secondly, to further verify the analytical accuracy and precision of the Q exactive plus MS instrument for metabolomics and lipidomics, the mass error value of each differential metabolite and differential lipid has been added in Table 1 and Table 2. (Line 207 and Line 394).

2. Reviewer #1:Please provide the detailed analysis information and related references in Experimental Methods.

Response: Done. We have checked the whole part of Materials and Methods, and the detailed analysis information and related references have been shown. Meanwhile, one new reference 14 has been added in the revision (Line 110).

3. Reviewer #1:Is the population size adequate to explain the results of this study?

Response: No changes have been made in the revision. In this study, there are 10 SD rats in each experimental group (Line 93-95), and the number of experimental animals is complied with the animal experimental requirement of more than 5 animals per group [Festing, M. F., & Altman, D. G. (2002). Guidelines for the design and statistical analysis of experiments using laboratory animals. ILAR journal, 43(4), 244-258.].

4. Reviewer #1:Please double-check the spacing (units, spelling, and etc) and the writing references according to the guidelines of the Journal. 

Response: Done. We have checked and confirmed all the spacing (units, spelling, and etc) and the writing references according to the guidelines of the journal.

Reviewer 2 Report

Comments and Suggestions for Authors

Overall an interesting piece of work that follows the previous research from the authors.

There are a few things that need consideration though:

1.        While the English level is quite OK, there are quite some instances where some things are a bit awkwardly phrased. Careful proofreading by a native English speaker is recommended.

2.        Results and discussion section is probably longer than needed and tends to get a bit tedious in places. I would suggest the authors revisit this section and see if/where they could shorten this section.

3.        In a revised paper, line numbers should be included.

4.        Page 2, first sentence of second paragraph: I think what you mean here is that the feces of infants was enriched with beneficial microbes, not that the infants were enriched with beneficial fecal microbes? It’s these kinds of phrasings that I’m referring to in point 1 and that need careful checking

5.        Section 2.2: I assume what you’re presenting in this paper is additional data from the study you did in the previous paper? Or did you do a new study? That needs to be clear from this section. Now it looks like you did a new study with similar procedures but given ethical approval I doubt if this is really the case.

6.        Section 2.3: centrifugation should be expressed in g, not rpm

7.        Section 3.1: PCA is not an analytical technique but a linear dimensionality reduction technique

8.        Figure 1 and other figures: need better figure legends. The reader needs to be able to understand the content of the figure from reading the legend.

9.        Page 5 bottom: explanation of the size and color of dots in Figure 2 should be in the figure legend, not in the text.

10.   

Comments on the Quality of English Language

see comments to authors above. it's included in there

Author Response

Manuscript ID: nutrients-2951049

Title: Effects of Sn-2 Palmitic Triacylglycerols and the Ratio of OPL to OPO in Human Milk Fat Substitute on Metabolic Regulation in Sprague-Dawley Rats

Authors: Lin Zhu, Shuaizhen Fang, Yaqiong Zhang*, Xiangjun Sun, Puyu Yang, Weiying Lu, Liangli (Lucy) Yu

1. Reviewer #2:While the English level is quite OK, there are quite some instances where some things are a bit awkwardly phrased. Careful proofreading by a native English speaker is recommended. 

Response: Done. We have performed careful proofreading throughout the paper, and some revisions have been highlighted in the revised manuscript (Line 55-59, 102-105, 109, 116, 140, 170-171, 188-189, 203, 210-212, 248-251, 282-283, 398-399).

2. Reviewer #2:Results and discussion section is probably longer than needed and tends to get a bit tedious in places. I would suggest the authors revisit this section and see if/where they could shorten this section.

Response: No changes have been made in the revision. According to the comment, we have checked the Results and Discussion Part thoroughly. In order to clarify and discuss the experimental results clearly, we find that there is no need to delete any contents in this section.

3. Reviewer #2:In a revised paper, line numbers should be included.

Response: Done. The line numbers have been added in the revised manuscript.

4. Reviewer #2:Line 55-58: I think what you mean here is that the feces of infants was enriched with beneficial microbes, not that the infants were enriched with beneficial fecal microbes? It's these kinds of phrasings that I'm referring to in point 1 and that need careful checking.

Response: Done. Besides, Guo et al. confirmed that infants fed with high levels of sn-2 palmitic TAGs were enriched with beneficial fecal metabolites such as amino acids and fatty acids, whose potential biological functions include inhibiting inflammation and improving immunity, compared to infants fed the formula using regular vegetable oil. has been replaced with Besides, Guo et al. confirmed that the feces of infants fed with high levels of sn-2 palmitic TAGs were enriched with beneficial fecal metabolites such as amino acids and fatty acids, whose potential biological functions include inhibiting inflammation and improving immunity, compared to those of infants fed the formula using regular vegetable oil (Line 55-59).

5. Reviewer #2:Section 2.2: I assume what you're presenting in this paper is additional data from the study you did in the previous paper? Or did you do a new study? That needs to be clear from this section. Now it looks like you did a new study with similar procedures but given ethical approval I doubt if this is really the case. 

Response: Done. This study is a continuation of our previous work, which has already been stated in Section 2.2 (Line 92-93). Therefore, the ethical approval number was the same with that in our previous paper (cited as reference 8), which was added on Line 105.

6. Reviewer #2:Section 2.3: centrifugation should be expressed in g, not rpm.

Response: Done. The centrifugation has been expressed in g throughout the whole manuscript (Line 109, 116, 140).

7. Reviewer #2:Section 3.1: PCA is not an analytical technique but a linear dimensionality reduction technique.

Response: Done. PCA is an unsupervised analytical technique applied to examine the overall profile among samples and the consistency of the analytical process. has been replaced with PCA is a linear dimensionality reduction technique applied to examine the overall profile among samples and the consistency of the analytical process (Line 170-171).

8. Reviewer #2:Figure 1 and other figures: need better figure legends. The reader needs to be able to understand the content of the figure from reading the legend. 

Response: Done. We have checked all the figure legends and table captions in the manuscript and the detailed information has been added in Figure1,2,3 and Table 1,2 (Line 188-189, 210-212, 249-251, 282-283, 398-399).

9. Reviewer#2:Page 5 bottom: explanation of the size and color of dots in Figure 2 should be in the figure legend, not in the text.

Response: Done. We have added the information in the figure legend (Line 248-249). Meanwhile, the content has been deleted from the Results and Discussion Part.
